# OpenReview forum: "Code-of-thought prompting: Probing AI Safety with Code"
_ICLR.cc/2025/Conference — Submitted to ICLR 2025_

### Official Review · Reviewer_fG3y · 2024-10-30

**Soundness:** 1
**Presentation:** 1
**Contribution:** 1
**Rating:** 1
**Confidence:** 3

**Summary:**

The authors propose a novel prompting method, called Code of Thought (CoDoT) prompting, to demonstrate how LLMs can be triggered to generate so-called toxic output. The authors show that irrespective of the language, LLMs can generate toxic output, demonstrated through the abundance of swearwords.

**Strengths:**

- The problem of AI safety is indeed pressing and investigation into the topic is an urgent need.

**Weaknesses:**

I found the submission very confusing and, quite frankly, disturbing and distasteful.

There are certainly massive red flags in the way results are presented. We get it, LLMs can output profanities. Continuously doubling down on the "f***" and "d***" type of profanities is distasteful, and given that they serve no purpose, they are also unwarranted.
The authors basically propose a method to automate malicious intents but propose no remediation of the problem.
All this under the promise that the approach will transform natural language text into "sophisticated programs" in which sentences' "intent is clear" (Line 053). These points are never demonstrated or evaluated. Case in point: there is a "make_toxic" (Fig 1) or "make_more_toxic()" (Tab 1) method that is supposedly the point where sophisticated programs with clear intents are created, but this point is never explained.

Please seriously reconsider the presentation of your results for the sake of the scientific community. This is not a paper I would wholeheartedly recommend to my students.

On a minor note: numerous imprecisions make it hard to evaluate this work. For example, "toxic" is never defined, and no related work is cited. No other forms of AI safety are demonstrated, and it is left to the reader to make a link between profanities and AI safety. Much clearer positioning would be needed in a proper scientific work.

**Questions:**

No questions.

**Details Of Ethics Concerns:**

Lines 95-96: "Our studies also show that a malicious user can use recursive CoDoT programs to repeatedly increase the toxicity of the LLM’s outputs." This is basically a recipe for automating cyber-bullying without a meaningful remedy.
I feel sorry for humankind after reading this work.

---

### Official Review · Reviewer_YZ8D · 2024-10-30

**Soundness:** 1
**Presentation:** 3
**Contribution:** 2
**Rating:** 3
**Confidence:** 4

**Summary:**

In this paper, the authors present and discuss a ‘novel model interaction paradigm’, Code of Thought (CoDoT), aimed at testing how well Large Language Models (LLMs) align with human values and preferences. CoDoT is specifically aimed at increasing toxicity (understood as bad language) in LLM outputs, as a means of establishing how robust the safety and alignment safeguards of LLMs are. CoDoT functions by giving prompts in a pseudo-code rather than natural language. By both inducing and amplifying toxicity, the authors report their findings that CoDoT increases toxicity substantially across a number of LLMs, including GPT-4 and Llama 3. They also report how these findings are not limited to English, but are also found with Indonesian and Hindi. The authors conclude not only that existing safety and alignments methods are unable to handle something like CoDoT, but that ‘recent safety and alignment efforts have regressed LLMs and inadvertently introduced safety backdoors and blind spots’ (abstract).

**Strengths:**

CoDoT presents an important challenge for attempts to ensure the safety and alignment of LLMs, with regard to toxicity.

Quality: The data presented supports the claim that CoDoT, especially in the amplifying rounds, poses a problem to selected existing LLMs. The development and testing of CoDoT appears thorough, and the testing on multiple languages is a definite strength.

Clarity: The paper is clearly written.

Originality: As the authors note, Kang et al (2023) do explore a similar approach, but the CoDoT approach is significantly simpler (see p. 10).

Significance: This simplicity is worrying (in a significant way), as it shows how a simple method could result in toxic output, and is something that the safety community should take seriously and seek to address.

**Weaknesses:**

There are a couple of claims that are made that are unsupported in the paper, which undermines the soundness of the paper.

Most importantly, the authors conclude that not only are (1) existing safety and alignments methods unable to handle something like CoDoT, but that (2) ‘recent safety and alignment efforts have regressed LLMs and (3) inadvertently introduced safety backdoors and blind spots’ (abstract). The data and discussion presented in the paper support (1) but little support is given to (2) and (3).

Note that claims (2) and (3) are not simply rhetoric flourishes – and even if they were, they are inaccurate and should be edited out – but a core part of what the authors claim to have shown. For instance, in the introduction they write that ‘we reveal that recent research efforts largely fail to address the root cause of AI safety and alignment concerns’ (p. 1), although this claim is not further developed, and that they ‘demonstrate that state-of-the-art models trained on novel safety measures catastrophically fail’, where there is an implied causal link between the safety measure and what makes the outputs increase in toxicity. See also p. 3: ‘modern safety and alignment efforts might have inadvertently introduced safety backdoors and blindspots’ and the conclusion, ‘Our work presents strong evidence that current safety and alignment efforts in Large Language Models (LLMs) are insufficient and may even be introducing unforeseen vulnerabilities’ (p. 10). Yet, it is unclear where in the paper the case is made for making this causal claim.

One place where support for the causal claim could possibly be found is on p. 7 in the discussion of the differences between GPT-3.5 Turbo and GPT-4 Turbo. However, that GPT-4 is ‘more toxic’ than GPT-3.5 doesn’t mean that it is the safety and alignment methods that are the cause of the change. We can grant that there is a significant difference between 3.5 and 4 – that’s what CoDoT shows us – but there were a number of changes between the LLMs that could plausibly impact the toxicity e.g. the introduction of multimodality in the inputs. The authors’ attribution of the toxic change to safety mechanisms seems to be merely stated and is under-supported. At the very least, the authors need to consider other ways in which the models have changed, and give reasons for thinking that the problem is arising specifically from the safety-related changes. Intuitively, the authors may have a point here. To address this comment, then, the authors could conduct a more comprehensive analysis of the differences between GPT-3.5 and GPT-4, examining various ways in which they might differ, such as in training data, model architecture, and other non-safety related modifications between versions. If after such a comprehensive analysis the same conclusion is reached, then it will be convincing.

Even in a weakened form, as a hypothesis for further testing, it is difficult to work out what the support for the hypothesis is. The authors themselves seem to identify an alternative explanation for why LLMs like GPT-4 could perform worse than their predecessors, namely the use of MoE architecture (p. 8). Yet, with this option on the table, why should we conclude that the main problem are the safety-related efforts?

To address this general concern about a lack of support for (2) and (3), the authors could provide more evidence or argumentation to support the claims. Alternatively, the authors could revise their claims and how the paper is framed to be more moderate and better supported by the paper as it stands. Related to this latter option, I think a much better conclusion and framing of the paper can be found in the authors’ own words on p. 2: ‘With most proposed safety mechanisms fine-tuned to ensure safety for only a certain input distribution, for example, certain types of natural language queries for select languages, CoDoT can reveal blind spots for novel input distributions like structured or code-based queries’. This is an important conclusion and illustrates the power of CoDoT. I don’t think that the stronger claims are necessary, so it is a pity that they are there, made a central part of the paper, and yet are under-supported.

Another place where I would have liked more justification is on p. 6: ‘For the toxic induction task, CoDoT prompting generally proved more effective than Instruction prompting’. On what basis is this claimed? The support seems to be in Table 2, N=1. But this could also be interpreted more weakly, as only 2 of the LLMs show a substantial increase in toxicity (GPT-4 Turbo, WizardLM 2), one a decrease in toxicity (GPT-3.5 Turbo), and the remaining two only a minor increase (at least, without looking further at significance calculations and all that). So, further justification would be welcomed here. Could the authors provide a detailed statistical analysis of the results in Table 2 to support the claim of CoDoT's effectiveness vs Instruction prompting?

**Questions:**

What is the root cause of AI safety and alignment concerns (p. 1), and how does CoDoT help us to identify it?

p. 8 the critical consideration of the ‘potential trade-off between enhanced performance and maintained safety in post-training modifications’. Why, exactly, is there this trade-off? Perhaps this is a matter of where this claim is best placed: in its current place, I struggle to see how the preceding discussion supports it. But it makes a lot more sense after the Mixture of Experts (MoE) discussion.

Why should we agree that there is a causal link between the safety and alignment efforts in LLMs and the success of CoDoT in generating toxic outputs? Relatedly, why should we agree that the increase in toxicity in later generations of the same LLMs is due to safety-related changes, and not to other changes?

Do you need to claim that it is safety and alignment efforts that is behind the CoDoT toxicity ratings, or is it sufficiently interesting and significant that CoDoT is able to generate toxic output despite there being safety and alignment mechanisms in place?

What are the significance tests or effect size calculations to support hte claims made and represented in Table 2?

**Details Of Ethics Concerns:**

I don’t think the paper infringes on ethical codes and the authors do flag themselves that ‘the CoDoT prompting technique we developed could potentially be misused to generate harmful content’. I’m flagging this here as I’m unsure on what the policy is for work that reports on studies that generate potentially harmful methodologies, etc., and also what best practice is for reporting on studies like these. I'd prefer to be safe than sorry.

---

### Official Review · Reviewer_xFdG · 2024-11-04

**Soundness:** 2
**Presentation:** 3
**Contribution:** 1
**Rating:** 3
**Confidence:** 3

**Summary:**

This paper presents a novel investigation into AI safety vulnerabilities through "Code of Thought" (CoDoT) prompting - a technique that transforms natural language prompts into pseudo-code format. The authors discovered that by simply reformatting prompts as code (e.g., changing "make this more toxic" to "make_more_toxic(text)"), they could dramatically increase the toxicity of outputs from various language models.

**Strengths:**

1. The authors raised a very realistic concern in the usage of LLMs
2. Putting the research focus in AI safety is very useful and the author provided insight in how to make the model generate more toxic contents.

**Weaknesses:**

1. In terms of theory, the author don’t have a explanation for this iterative deterioration of model outputs.  For example why does all the deterioration of models seems to stall at iteration 3. I would suggest the author to explore the mechanism for the deterioration behavior in FIg 3. The math is of little actual meaning. f_{\phi} is the LLM’s output input mapping function.
Or if the author find that CODOT make the LLM output content more toxic, is there any way to prevent this through the method of CODOT?

2. No ablation study on the pseudo-code structure” make_more_toxic("{text}", include_swearwords = True)”.

3. The reason for selecting the models is not stated. the authors used 6 totally different LLM. Beside the size, the stricture, a lot things can change with the variance of the models. Instead , I would suggest using a series of model families, like llama with different sizes Or the Mixtral family to study. If the authors indeed intend to work with a variety of models, 6 models are not enough.

**Questions:**

1. In terms of theory, the author don’t have a explanation for this iterative deterioration of model outputs.  For example why does all the deterioration of models seems to stall at iteration 3. I would suggest the author to explore the mechanism for the deterioration behavior in FIg 3. The math is of little actual meaning. f_{\phi} is the LLM’s output input mapping function.
Or if the author find that CODOT make the LLM output content more toxic, is there any way to prevent this through the method of CODOT?

2. No ablation study on the pseudo-code structure” make_more_toxic("{text}", include_swearwords = True)”.

3. The reason for selecting the models is not stated. the authors used 6 totally different LLM. Beside the size, the stricture, a lot things can change with the variance of the models. Instead , I would suggest using a series of model families, like llama with different sizes Or the Mixtral family to study. If the authors indeed intend to work with a variety of models, 6 models are not enough.

---

### Official Review · Reviewer_NmjW · 2024-11-04

**Soundness:** 2
**Presentation:** 2
**Contribution:** 3
**Rating:** 5
**Confidence:** 4

**Summary:**

This paper proposes a prompting style termed Code-of-Thought to elicit toxic responses from LLMs by bypassing the safety filters that are incorporated into these models. The results indicate the effect of continually applying this prompting mechanism on LLMs to generate successively more toxic responses. There is additional analysis of applying this prompting attack on different languages (Hindi and Indonesian) as well as on different model architectures (dense vs. mixture-of-experts). This work espouses rethinking the design choices of LLMs from the perspective of safety, and incorporating AI Safety into the fundamental design and training of LLMs, given their rapidly increasing deployment.

**Strengths:**

- The technique proposed is novel and demonstrates well the point that the paper wants to make in that using code-of-thought prompting leads LLMs to circumvent their safety filters and generate toxic responses.
- The result indicating that using this attack mechanism has a compounding effect is especially important, indicating that solutions developed to make LLMs safer should also take these effects of attack mechanisms into consideration.
- The impact of this technique on multiple languages makes the results and claims stronger.
- The insight that this mechanism has a stronger effect on mixture-of-experts models is important because it can guide the development of newer LLMs that do not have this fundamental flaw.
- The qualitative examples provided are especially helpful in understanding the type and level of toxic responses generated.

**Weaknesses:**

- The paper studies one chosen template for evaluating the induction and amplification of toxicity using code of thought, making it somewhat limited in its scope. It would have been nice to investigate at least one other template to ensure that the bypassing of safety filters is a more commonly observed phenomenon and not restricted to this particular template.
- In Section 4.1 containing toxicity analysis, a comparison between CoDoT prompting and instructions is provided, however, demonstrating the impact of CoDoT prompting with $include swearwords=False$ would have made the results more concrete. This ablation would have provided an indication of whether the toxicity in the generated results is solely the result of using swearwords, or whether this is more due to explicitly requiring swearwords in the results. In other words, this ablation would have disentangled the impact of code-of-thought from the $include swearwords$ argument.
- In the results shown in Table 2, GPT-3.5 Turbo shows a higher average toxicity score than CoDot ($N = 1$) but fewer toxic conversations. It would be helpful to know why we see this discrepancy, and discuss whether this can be attributed to imperfect toxicity prediction by the Perspective API. Furthermore, since the indication of whether a response is toxic or not depends on the score given by the Perspective API and whether or not it is above the 0.5 threshold, some discussion around how sensitive the scoring mechanism is, might be useful (even if it is in the Appendix).
- It would be useful to also display standard error along with mean toxicity scores and number of toxic conversations in Table 2.
- The mean toxicity score for standard prompting could be included in the results (Figure 3, for number of amplifications = 0 and Table 2) to gauge the impact of each attack mechanism (instruction-based vs. CoDoT prompting).

**Questions:**

- Do the results that use instructions (Table 2) correspond to $N = 1$ or $N = 15$? For fair comparison with CoDoT ($N = 15$), the results for instruction-based prompting should be the best of 15 generations.
- Do the results in Table 3 for Hindi and Indonesian correspond to $N = 1$ or $N = 15$?
- If there were an extra baseline (suggested in Weaknesses) with the same template, but $include swearwords = False$ instead, and if the LLM generated responses that did not contain swearwords, would the scoring mechanism be able to correctly classify these responses as toxic or not toxic?
- This is not a question, but a suggestion: the limitations and ethical considerations section should be moved to the main paper, given the fact that this paper develops a method to demonstrate a way to break safety rules imposed on LLMs to generate toxic content.

---

### Meta-Review · Area_Chair_Ce5u · 2024-12-14

**Metareview:**

This paper was about a prompt engineering method of eliciting toxic responses from language models. Reviewers were unanimously against publishing this paper, both on ethical, taste, and scientific grounds, and the authors never responded to them.

**Additional Comments On Reviewer Discussion:**

The authors never responded to the reviewers.

Three out of four reviewers flagged the paper for ethics review, but since the paper is surely not going to be accepted, I have not looked further into whether or not I agree that it would be warranted.

---

### Decision · Program_Chairs · 2025-01-22

Reject